# Reference Gene Selection for qPCR Analysis in *Schima s**uperba* under Abiotic Stress

**DOI:** 10.3390/genes13101887

**Published:** 2022-10-18

**Authors:** Jun Yao, Gang Zhu, Dongcheng Liang, Boxiang He, Yingli Wang, Yanling Cai, Qian Zhang

**Affiliations:** 1Guangdong Provincial Key Laboratory of Silviculture Protection and Utilization, Guangdong Academy of Forestry, Guangzhou 510520, China; 2College of Horticulture and Landscape Architecture, Zhongkai University of Agriculture and Engineering, Guangzhou 510225, China

**Keywords:** *Schima*
*superba*, reference gene, abiotic stress, tissues

## Abstract

Quantitative real-time PCR (qPCR) is an indispensable technique for gene expression analysis in modern molecular biology. The selection and evaluation of suitable reference genes is a prerequisite for accurate gene expression analysis. *Schima*
*superba* is a valuable tree species that is environmentally adaptable and highly fire-resistant. In this study, 12 candidate reference genes were selected to check their stability of gene expression in different tissues under abiotic stresses: cold stress, salt stress, and drought stress by ΔCt, geNorm, NormFinder, BestKeeper, and RefFinder. The results indicated that *AP-2* was the most stably expressed overall and for the cold stress and drought stress. *eIF-5α* gene expression was the most stable under the salt stress treatment, while *UBQ* expression was the most stable across mature leaves, shoots, stems, and roots. In contrast, *UBC20*, *GAPDH*, and *TUB* were the least stably expressed genes tested. This study delivers valid reference genes in *S. superba* under the different experimental conditions, providing an important resource for the subsequent elucidation of the abiotic stress adaptation mechanisms and genes with biological importance.

## 1. Introduction

To date, quantitative real-time PCR (qPCR) has been widely used in many fields, such as agriculture, genetics, microbiology, and medical gene expression detection and molecular technology for quantitative research, based on its significant advantages in sensitivity, accuracy, specificity, and operability [1,2]. However, the accuracy of qPCR results is affected by the number of initial templates, RNA quality, reverse transcription efficiency, amplification efficiency, and reference genes [3]. In addition, the selection of reference genes is an important factor affecting the stability of qPCR results. In general, gene expression can be standardized or quantified by using one or more stable reference genes. Ideally, the internal reference genes should be stably expressed under different tissues, developmental stages, and abiotic stresses of the plant. However, many studies have demonstrated that plants lack a common reference gene. The suitability of specific reference genes depends on the special tissue or the experimental conditions [4]. Therefore, selecting and validating reference genes for specific experimental conditions is a necessary prerequisite for the reliability of qPCR results [5]. 

*S. superba* is a large evergreen broad-leaved tree in the Theaceae (Appendix A). Its timber is excellent, solid, and tight [6]. *S. superba* is the main fire prevention forest belt construction tree in the eastern subtropical region of China due to its high water content of fresh leaves, high fire point, and an oil content of only 6% [7]. At the same time, it is classified as a valuable tree species because of its high economic value [8]. *S. superba* is highly adaptable and can grow under unfavorable conditions such as acidic soil, barren mountains, and arid areas [9]. However, the mechanism regulating the adaptation of *S. superba* to abiotic stresses has not been reported. Elucidating the abiotic stress mechanism of *S. superba* and detecting the expression of related genes are important for the selection and breeding of superior varieties. The changes in gene expression levels induced by abiotic stresses such as drought, salinity, and cold can be complex and multifaceted, often affecting the expression levels of stable reference genes under other experimental conditions. To date, reference genes have been reported in *S. superba* [10], but these genes have only been validated under normal experimental conditions. Many stable reference genes under abiotic stresses have been reported from different plants, including rice [11], luffa [12], and sorghum [13]. The stable expression of reference genes in *S. superba* under abiotic conditions has not been reported. Therefore, the main objective of this study was to identify reference genes that exhibit high expression stability under various abiotic stress conditions in *S. superba* to facilitate subsequent studies on abiotic stress mechanisms and to provide an important research basis for the selection and breeding of superior varieties.

## 2. Materials and Methods

### 2.1. Plant Materials and Treatments

Plant materials were collected from the nursery of the Guangdong Academy of Forestry. Seedlings were selected from 1–2 years of age and cultivated in artificial climate incubators with a 16-h artificial light–8-h dark cycle and 65–75% relative humidity. For the cold stress treatment, seedlings were grown at 16 °C, and 25 °C was selected for the other stress treatments. For salt stress treatment, seedlings were treated with 200 mM NaCl. For the drought stress treatment, seedlings were watered with the 20% PEG 6000 solution. Leaves were then collected at 0, 1, 3, 6, 9, 12, 24 h, and 3 d after the treatments. Plant tissues were collected from mature leaves, shoots, stems, and roots. Each experiment was completed with three replicates and immediately frozen in liquid nitrogen and stored at −80 °C until use.

### 2.2. Reference Gene Selection and Primer Design

Candidate genes were selected based on pre-lab transcriptomic data from our laboratory (unpublished) and conventional reference genes. The primers for qPCR were designed using primer 5.0 (Appendix A) and synthesized by Tsingke Biotechnology Co., Ltd. (Beijing, China).

### 2.3. qPCR Analysis

The total RNA was extracted using the RNAprep Pure Plant Plus Kit (polysaccharides and polyphenolics-rich) (Code No. DP441, TIANGEN, Beijing, China). RNA quality and concentration were assessed by 1% agarose gel electrophoresis and the BioDrop nucleic acid protein analyzer. The RNA samples with a 260/280 nm absorbance ratio of 1.8–2.0 were used to synthesize the first strand cDNA with the PrimeScript™ RT reagent Kit with gDNA Eraser (Perfect Real Time) (Code No. RR047A, TAKARA, Beijing, China) for further analyses.

qPCR follows the guidelines of the Biomarker 2 × SYBR Green Fast qPCR Mix (Code No. RK02001, BioMarker, Beijing, China): 8.0 μL of Nuclease-free water, 10 μL of 2 × SYBR Green Fast qPCR MIX, 0.5 μL of forward and reverse primers (10 μM), and 1 μL of diluted cDNA. qPCR was conducted using BIO-RAD CFX Connect (Bio-Rad Laboratories, Hercules, California, CA, USA) with the following cycling conditions: initial denaturation at 95 °C for 3 min followed by 40 cycles of 95 °C for 5 s, 60 °C for 30 s, and 72 °C for 20 s. The relative quantification in each sample was determined. A blank control with double-distilled water as a template was also analyzed, and three independent biological replicates and three technical repetitions were performed for each of the quantitative PCR experiments. The cDNA samples were diluted in a 10-fold gradient to measure the threshold cycle (Ct) (10,000-fold template concentration was too low for some candidate genes to determine the exact Ct value), and the standard curve was plotted using Excel with the horizontal coordinate as the dilution and the vertical coordinate as the mean Ct value. The linear slope K and correlation coefficient R^2^ of the candidate reference genes were analyzed, as well as the amplification efficiency E, calculated as E = (10^−1/K−1^) × 100%.

### 2.4. Statistical Data Analysis

The qPCR cycle threshold (Ct) value was recorded for each candidate reference gene under different treatments. The ΔCt [14], geNorm [15], NormFinder [16], BestKeeper [17], and RefFinder [18] algorithms were used to assess the expression stability of candidate reference genes. The Bestkeeper and ΔCt algorithms assess the stability of candidate reference genes based on Ct values, whereas the geNorm and NormFinder algorithms are based on the 2^−ΔCt^ values obtained from the Ct value transformation. Finally, the reference genes were ranked based on the geometric mean (GM) values calculated by RefFinder.

## 3. Results

### 3.1. Candidate Reference Genes and PCR Amplification

The genes selected as potential reference genes were actin (*Actin*), glyceraldehyde-3-phosphate dehydrogenase (*GAPDH*), ubiquitin (*UBQ*), ubiquitin-conjugating enzyme (*UBC*), α tubulin (*TUA*), eukaryotic initiation factor 5A (*eIF5α*), ribosomal protein L17 (*RPL17*), elongation factor-1α (*EF1α*), AP-2 complex subunit mu-like (*AP-2*), UDP-galactose transporter (*UDP*), tubulin β chain (*TUB*), GIIaglucosidase II α-subunit (*GIIα*). These genes were selected as stable reference genes in other species [19,20,21,22,23,24,25].

The primer specificities were confirmed by 1% gel electrophoresis and melting curve analyses. All primers amplified a single amplicon of the expected size (Appendix A). The candidate reference genes were analyzed and the solubilization temperatures of the candidate reference genes’ melting curves were consistent, while the candidate reference genes showed a single specific peak (Appendix A). The results indicated that the 12 primer pairs were highly specific.

The general amplification efficiency should be in the range of 90–120%. The highest amplification efficiency of the 12 candidate reference genes was 123.13% and the lowest was 92.35%. The correlation coefficients R^2^ of all candidate genes were above 0.98 (Table 1). Therefore, the linearity and amplification efficiency of the candidate reference genes were largely satisfied by qPCR analysis.

### 3.2. Ct Values of Candidate Reference Genes

The Ct values of the candidate reference genes ranged from 21.20 to 34.61 under different experimental treatments, and the Ct values were inversely proportional to the gene expression abundance. As shown in Figure 1, The box plots reflect the differences in the distribution of Ct values among the different candidate reference genes, and the dispersion degree indicates the stability of genes. A lower dispersion degree indicates more stable gene expression with the same experimental sample. *UBQ* has the highest concentration trend, and *TUB* has the lowest concentration trend.

### 3.3. Stability of Candidate Reference Genes

The ΔCt algorithm is based on the mean standard deviation (mSD). The level of the mSD value indicates the stability of the internal reference gene, and the smaller the mSD value indicates the higher the stability of the internal reference gene. In this study, the most stably expressed genes were different under different experimental conditions (Figure 2). For all the experimental conditions, *AP-2*, *UBC4*, and *UBQ* were the most stably expressed genes. Under cold stress, *AP-2*, *eIF-5α*, and *UDP* were the most stably expressed genes. For the salt stress, *eIF-5α*, *AP-2*, and *UBC4* were the most stably expressed genes. For the drought stress, *eIF-5α*, *Actin*, and *UBC4* were the most stably expressed genes. For the different tissues, *UBQ*, *EF1α*, and *UBC4* were the most stably expressed genes. *GAPDH*, *UBC20*, and *TUB* showed unstable expression under all experimental conditions.

The geNorm algorithm ranks the stability of reference gene expression based on the calculation of the average M value. Larger M values for candidate reference genes indicate lower stability (M value threshold of 1.5). In this study, the most stably expressed genes were different under different experimental conditions (Figure 3). For all the experimental conditions, *AP-2* and *Actin* expression were the most stable with an M value of 0.38. For the cold stress, *AP-2* and *eIF-5α* expression were the most stable with an M value of 0.33. For the salt stress, *AP-2* and *eIF-5α* expression were the most stable with an M value of 0.34. For the drought stress, *AP-2* and *Actin* expression were the most stable with an M value of 0.2. For the different tissues, *UBQ* and *EF1α* expression were the most stable with an M value of 0.13. Consistent with the ΔCt algorithm, *GAPDH*, *UBC20*, and *TUB* expressions showed the most unstable performance.

The pairwise variations value (V) is the value of pairwise variations of the standardized factor. The default threshold value of V is 0.15. If the value of Vn/Vn + 1 is less than 0.15, then n is the optimal number of internal genes, and if the value of Vn/Vn + 1 is greater than 0.15, then n + 1 is the optimal number of reference genes. As shown in Figure 4, for all the experimental conditions, V2/V3 values were equal to 0.15 and V3/V4 were equal to 0.11, less than 0.15, so three reference genes are sufficient for normalizing gene expression data. In cold stress, salt stress, drought stress, and different tissues, V2/V3 was less than 0.15, demonstrating that two reference genes were sufficient to normalize the expression of the target gene.

NormFinder assesses expression stability (S value) by the variance method. Lower S values correspond to higher gene expression stability. As shown in Table 2, for all the experimental conditions, *AP-2* expression was the most stable with an S value of 0.087. Similarly, the stability of *AP-2* expression was highest under the experimental conditions of drought and salt stress. Under cold stress, *UDP* has the best stability. For the different tissues, the expression of *UBQ* and *EF1α* was equally stable, whereas *TUB* was the least stably expressed reference gene.

The BestKeeper algorithm is used to assess the stability of the expression of the reference genes by calculating the standard deviation (SD) and the variation deviation (CV). High SD and CV values indicate low gene stability. As shown in Table 3, in all samples, the three most stably expressed genes were *UBQ*, *EF1α*, and *UBC4*. In cold stress, the most stably expressed genes were *UBQ*, *EF1α*, and *eIF-5α*. In drought stress, the most stably expressed genes were *UBQ*, *UBC4*, and *AP-2*. In salt stress, the most stably expressed genes were *UBQ*, *eIF-5α*, and *RPL17*. In different tissues, the most stably expressed genes were *RPL17*, *GIIα*, and *eIF-5α*. It is noteworthy that the *RPL17* gene was assessed to be more stable in the BestKeeper algorithm and less stable in the NormFinder algorithm under different tissue and salt stress experimental conditions.

The results of the ΔCt, geNorm, NormFinder, and BestKeeper algorithms were combined and ranked comprehensively using RefFinder (Figure 5). The results showed that *AP-2* was the most stably expressed reference gene in all samples, also under cold stress and drought stress experimental conditions. For the salt stress, *eIF-5α* expression was the most stable, while *UBQ* expression was the most stable in different tissues. In contrast, *UBC20*, *GAPDH*, and *TUB* were the least stably expressed genes.

## 4. Discussion

The stability of reference genes can directly affect the accuracy and stability of qPCR results [26]. The selection of an appropriate reference gene is the key to target gene expression studies. However, plants lack a common reference gene, and the reference genes are relatively stable only under specific tissue or experimental conditions. The same reference genes show different stability in different species. For example, *Actin* showed the best stability in *Prunus persica* [27] and Pitaya [28], but the worst in *Citrus sinensis* [29] and *Populus* [30], and *EF1α* had superior stability in blueberry [31], but not in apple [32]. The stability of the same reference gene also differs in different experimental conditions and tissues, and the expression of the reference gene is influenced by biotic and abiotic factors. For example, *EF1α* showed the highest stability in ‘Xiacui’ samples while being the most unstable gene among all samples of *Prunus persica* [27]. Gene expression patterns in plants are more complex and diverse under abiotic stress. Therefore, it is necessary to verify the stability of reference genes under different experimental conditions. Meanwhile, the selection of suitable candidate genes also has an impact on the results. The published studies on the reference genes of *S. superba* mainly focus on the stability of the expression of reference genes in different tissues under normal experimental conditions [10], and the candidate reference genes such as *Actin*, *eIF5α*, *GAPDH*, and *TUB* were selected, and their comprehensive evaluation of expression stability was similar to this study. *Actin* is more stable than *eIF5α*, while the worst stability was found in *GAPDH* and *TUB*. Unlike in the present study, the common candidate genes UBQ and EF1α were selected among the candidate genes, which had superior stability in different tissues. This study also investigated the expression levels of more complex abiotic stress conditions.

In this study, the stability of 12 candidate internal reference genes (*Actin*, *AP-2*, *EF1α*, *eIF-5α*, *GAPDH*, *GIIα*, *RPL17*, *TUB*, *UBC20*, *UBC4*, *UBQ*, *UDP*) was analyzed under four experimental conditions: cold stress, drought stress, salt stress, and different tissues. The gene expression stability was evaluated by the ΔCt, geNorm, NormFinder, BestKeeper, and RefFinder algorithms. The data showed that *AP-2*, *UBQ*, and *Actin* were the most stable internal reference genes of all samples, and *Actin*’s performance in different tissues of plants is similar to Yang’s study [10]. It is worth noting that *AP-2* performs well in all three types of abiotic stresses and its expression is, relatively, not the most stable in different tissues, which may indicate that *AP-2* is able to maintain stable expression, especially under complex abiotic stresses. *UBQ* is able to maintain a relatively stable expression, especially in different organizations. In contrast, *Actin* expression stability fluctuates under different experimental treatments, phenotypes are relatively unstable under cold treatments, and phenotypes are not optimally stable under other treatments. For the salt stress, *eIF-5α* expression was the most stable, while *UBQ* expression was the most stable in different tissues. These stably expressed genes are also used as suitable reference genes in other plants. *AP-2* (adaptor protein-2 complex) was identified as a potential reference gene due to stability in *Arabidopsis* [33] and grapevine [34]. The expression of *AP-2* was also stable under abiotic stresses in *Clerodendrum trichotomum* [23] and *Sedum alfredii* [35]. *eIF-5α* (eukaryotic translation initiation factor 5A) exhibited highly stable expression by microarray analysis of Arabidopsis [33]. Stable expression has also been demonstrated in some species, such as Tree Peony [25] and *Metasequoia* [36]. *UBQ* (ubiquitin family 6) is found in all eukaryotes and is highly conserved in the amino acid sequence. Gene expression levels are well stabilized in different tissues, such as Chinese prickly ash [37] and *Populus* [38]. A ubiquitin tag is reported to mark particular proteins for proteolytic elimination, but it can also have nonproteolytic functions [39]. Thus, its wide range of functions leads to the variable expression of *UBQ* in different plants, such as *Passiflora edulis* [40]. Other candidate genes: *Acti**n* [19], *EF1α* [20], *GAPDH* [41,42], *GIIα* [21], *RPL17* [43], *TUB* [44], *UBC* [22], and *UDP* [24] were selected as stable reference genes in other species, but not the most stable in this study. Normalization of the expression data of a target gene using one or more stably expressed reference genes is generally performed. Applying multiple genes may increase the accuracy and reliability of data normalization to some extent [45]. However, combined with the geNorm analysis results in this study, we believe that the use of two reference genes can better ensure the stability of the experiment and the accuracy of the results. The stably expressed reference genes obtained in this study will contribute to the study of gene expression levels in *S. superba* under abiotic stress, facilitate the study of abiotic stress mechanisms, and help discover new genes and signaling networks used by *S. superba* to cope with these challenges, which is essential for the development of new varieties with enhanced tolerance to stress.

## Figures and Tables

**Figure 1 genes-13-01887-f001:**
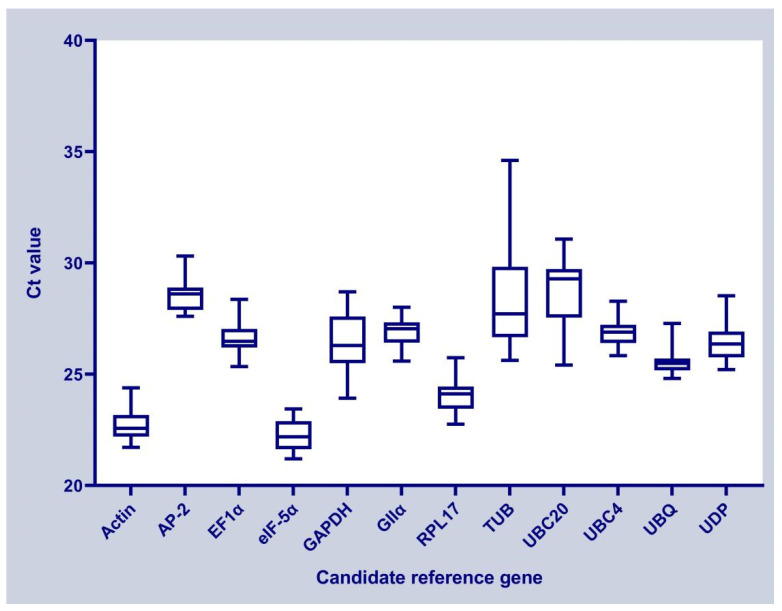
Distribution of Ct values among the 12 candidate reference genes.

**Figure 2 genes-13-01887-f002:**
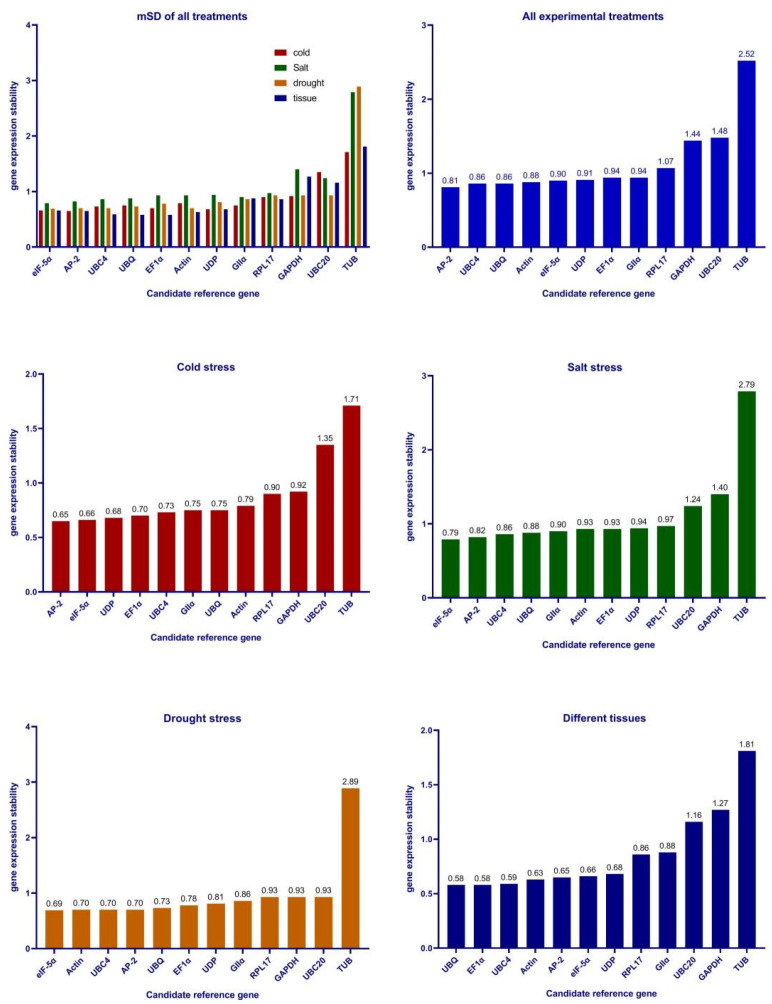
ΔCt stability analysis of 12 candidate internal reference genes.

**Figure 3 genes-13-01887-f003:**
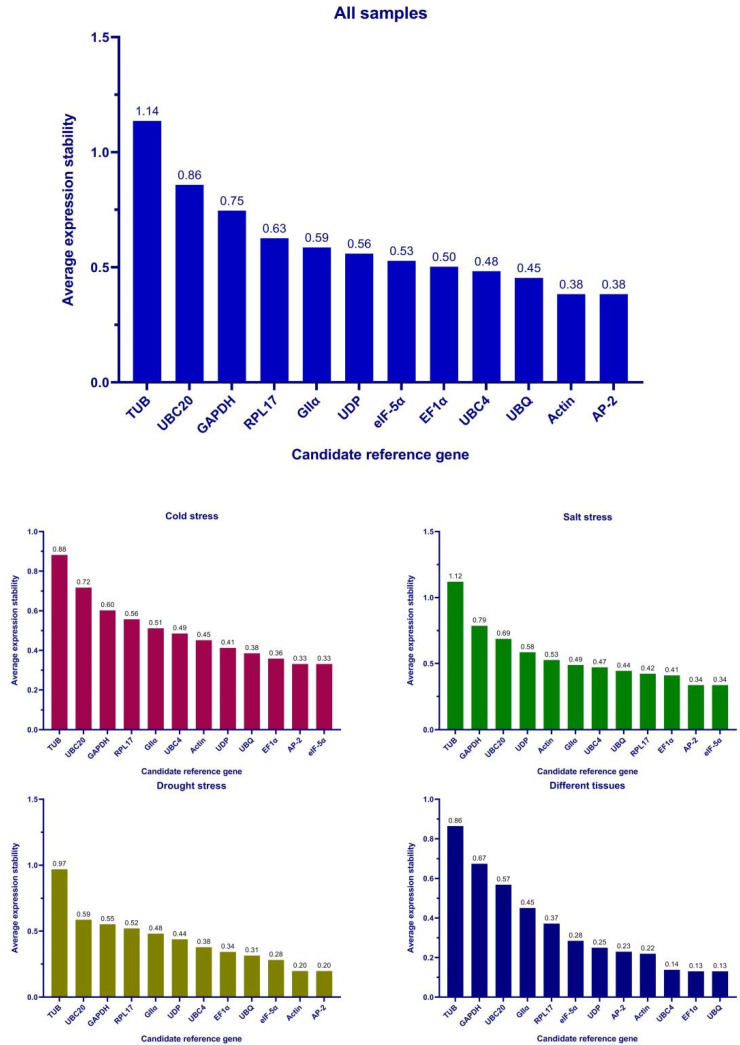
Expression stability of twelve reference genes under different conditions based on a geNorm analysis.

**Figure 4 genes-13-01887-f004:**
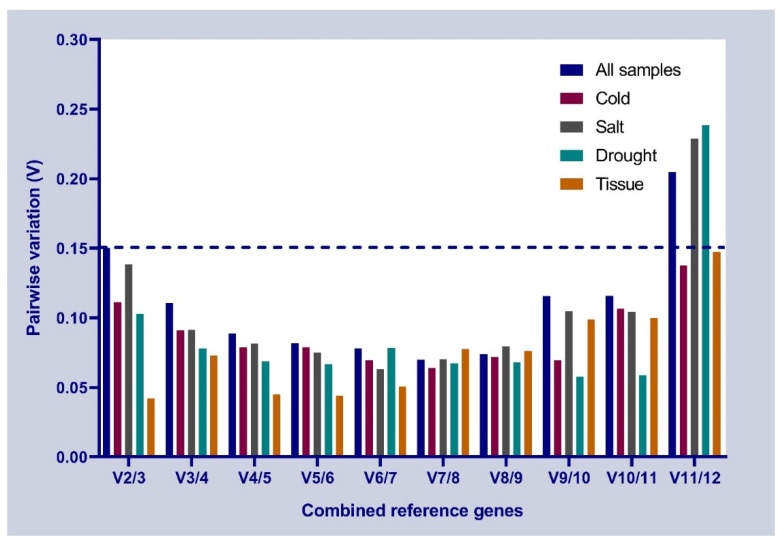
Pairwise variations value under different conditions based on a geNorm analysis.

**Figure 5 genes-13-01887-f005:**
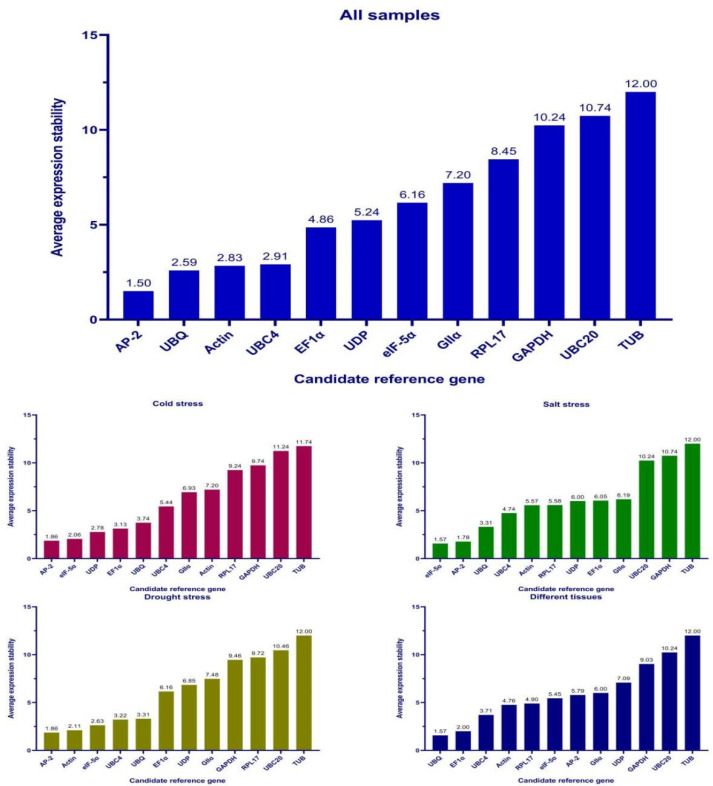
Comprehensive ranking of stability.

**Table 1 genes-13-01887-t001:** Amplification efficiency and correlation coefficient of candidate reference genes.

Reference Genes	Slope (K)	R^2^	Amplification Efficiency (E)
*UBQ*	−2.8669	0.9987	123.13%
*AP−2*	−2.9375	0.9875	118.99%
*Gllα*	−2.9570	0.9943	117.86%
*UBC4*	−2.9876	0.9942	116.13%
*elF−5α*	−3.1023	0.9938	110.06%
*TUB*	−3.1214	0.9950	109.11%
*GAPDH*	−3.1213	0.9996	109.11%
*RPL17*	−3.1270	0.9985	108.83%
*UDP*	−3.1845	0.9833	106.07%
*EF1α*	−3.1863	0.9998	105.99%
*Actin*	−3.4885	0.9953	93.49%
*UBC20*	−3.5201	0.9930	92.35%

**Table 2 genes-13-01887-t002:** Stability analysis of candidate reference genes based on NormFinder algorithm.

Rank	All Samples		Cold		Drought		Salt		Tissue	
	Gene	SV	Gene	SV	Gene	SV	Gene	SV	Gene	SV
1	*AP-2*	0.087	*UDP*	0.171	*AP-2*	0.069	*AP-2*	0.218	*UBQ*	0.066
2	*UDP*	0.251	*AP-2*	0.183	*Actin*	0.086	*UDP*	0.228	*EF1α*	0.066
3	*UBC4*	0.305	*eIF-5α*	0.221	*UBC4*	0.176	*eIF-5α*	0.273	*UBC4*	0.067
4	*Actin*	0.330	*EF1α*	0.309	*eIF-5α*	0.233	*UBC4*	0.392	*Actin*	0.182
5	*UBQ*	0.368	*UBC4*	0.323	*UDP*	0.292	*Actin*	0.421	*AP-2*	0.237
6	*eIF-5α*	0.421	*GIIα*	0.398	*UBQ*	0.297	*UBQ*	0.513	*UDP*	0.271
7	*GIIα*	0.477	*UBQ*	0.399	*GIIα*	0.462	*GIIα*	0.522	*eIF-5α*	0.307
8	*EF1α*	0.551	*Actin*	0.466	*EF1α*	0.518	*EF1α*	0.612	*GIIα*	0.635
9	*RPL17*	0.762	*GAPDH*	0.671	*UBC20*	0.579	*RPL17*	0.727	*RPL17*	0.668
10	*UBC20*	1.223	*RPL17*	0.678	*GAPDH*	0.694	*UBC20*	0.837	*UBC20*	0.953
11	*GAPDH*	1.246	*UBC20*	1.254	*RPL17*	0.755	*GAPDH*	1.175	*GAPDH*	1.176
12	*TUB*	2.448	*TUB*	1.641	*TUB*	2.861	*TUB*	2.741	*TUB*	1.760

**Table 3 genes-13-01887-t003:** Stability analysis of candidate reference genes based on BestKeeper algorithm.

Rank	All Samples			Cold			Drought			Salt			Tissue		
	Gene	SD	CV	Gene	SD	CV	Gene	SD	CV	Gene	SD	CV	Gene	SD	CV
1	*UBQ*	0.31	1.20	*UBQ*	0.14	0.53	*UBQ*	0.16	0.65	*UBQ*	0.21	0.80	*RPL17*	0.40	1.64
2	*EF1α*	0.45	1.70	*EF1α*	0.31	1.14	*UBC4*	0.26	0.98	*eIF-5α*	0.27	1.17	*GIIα*	0.67	2.54
3	*UBC4*	0.48	1.78	*eIF-5α*	0.42	1.94	*AP-2*	0.34	1.19	*RPL17*	0.33	1.38	*eIF-5α*	0.68	3.06
4	*Actin*	0.50	2.19	*UDP*	0.49	1.87	*eIF-5α*	0.34	1.56	*Actin*	0.35	1.54	*EF1α*	0.72	2.66
5	*AP-2*	0.51	1.79	*UBC4*	0.49	1.80	*Actin*	0.34	1.51	*AP-2*	0.37	1.30	*GAPDH*	0.73	2.88
6	*GIIα*	0.53	1.99	*AP-2*	0.51	1.80	*EF1α*	0.36	1.36	*GIIα*	0.40	1.46	*UBQ*	0.78	3.01
7	*RPL17*	0.54	2.26	*Actin*	0.54	2.39	*GIIα*	0.39	1.43	*UBC4*	0.42	1.54	*UBC4*	0.78	2.90
8	*eIF-5α*	0.58	2.62	*GIIα*	0.58	2.18	*GAPDH*	0.44	1.72	*EF1α*	0.47	1.77	*Actin*	0.94	4.14
9	*UDP*	0.60	2.28	*RPL17*	0.76	3.19	*UDP*	0.47	1.76	*UDP*	0.67	2.51	*AP-2*	0.95	3.30
10	*GAPDH*	1.02	3.86	*GAPDH*	0.80	2.93	*RPL17*	0.50	2.07	*GAPDH*	0.94	3.47	*UDP*	1.04	3.89
11	*UBC20*	1.18	4.10	*TUB*	1.03	3.77	*UBC20*	0.65	2.19	*UBC20*	0.97	3.33	*UBC20*	1.11	4.03
12	*TUB*	1.97	6.90	*UBC20*	1.28	4.55	*TUB*	2.33	8.05	*TUB*	2.25	7.87	*TUB*	1.95	6.48

## Data Availability

Not applicable.

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
