# Peer review of "Reference Gene Selection for qPCR Analysis in Schima superba under Abiotic Stress"

_genes, 2022, doi:10.3390/genes13101887_

Round 1
Reviewer 1 Report
The authors represented the evaluation of candidate reference genes in Schima sperba for valid quantitative gene expression analyses. A total of 12 candidate reference genes were evaluated for the gene expression stability in the three-types of abiotic stress samples of mature leaves, shoots, stems, and roots. Their results represented that AP-2 is the best reference gene among them. It should be useful for abiotic stress studies on Schima sperba. For making a better manuscript, comments were left below.
The reviewer thinks that text writing and data representation can be improved. (1) The technical English quality has to be raised up by an extensive proofreading by a native biologist. Grammatical errors including the issues of “italic or non-italic” and “capital or letter” and misuse of vocabulary were frequently found. (2) Software tools used in this study have to be referred by listing the original papers. (3) A subsection for the selection of potential reference genes should be inserted into the beginning of “Results” instead of in “Materials and Methods” section. The reasons for each potential reference gene should be described using references and/or commonly acceptable thought in biology. (4) Basic primer information (Table 1, Figure 1, and Figure 2) are minor technical issues and don’t contain biological contexts. Hence these Table and Figures could be moved to “Supplementary Materials”. (5) Most of readers of “genes” are unfamiliar with Schima sperba, hence it can be fine to represent photographs of this species samples (normal appearance and samples under abiotic stress conditions) in a supplementary material if available. (6) Novelty of this study is not clearly shown: it is better to mention that no reference gene is available for Schima sperba, or no reference gene with any serious evaluation result is available for this species. (7) Information of experimental replicates are unknown. How many biological replicates and how many technical replicates were analyzed in qRT-PCR assays? (8) It should be good to refer similar studies on the discovery of reference genes under abiotic stress in multiple plant tissues (organs) in Introduction or Discussion sections. One of examples is Pabuayon et al (2016) “Reference genes for accurate gene expression analyses across different tissues, developmental stages and genotypes in rice for drought tolerance. Rice, 9, 32. doi: 10.1186/s12284-016-0104-7.”
The reviewer does not understand the importance of checking/showing the absolute Ct values of reference genes from the text. If this point is critical for the evaluation of reference genes, it has to be described clearly. One possible aspect might be that genes with very low-level expressions may not be suitable as references because dilutions of cDNAs are often conducted in qRT-PCR experiments, and over-dilution could lead to give high variations of Ct values.
One potential thing for improving the contents is about the data set design. The authors analyzed the gene expression stability in four types of data sets separately (cold stress, drought stress, salt stress, and multiple tissues). These data set designs are good to find good reference genes in a specific sample series while genes with stable expression patterns could be driven by particular promoters (constitutive ones perhaps). Hence, the increase of sample types in the same data set for gene expression stability analysis would provide a benefit to decrease a risk to dismiss special expression characteristics that will be appeared in other samples untested. It would be interesting to compare the current results and a result in the combined data set to discuss deeply.
Author Response
Dear reviewers,
Thanks very much for taking your time to review this manuscript. I really appreciate all your comments and suggestions! We have carefully considered the suggestion of reviewer and make some changes(Please see the attachment). We have tried our best to improve and made some changes in the manuscript. Please find my itemized response in below and my corrections in the re-submitted files.

Reviewer 2 Report
The present study, 12 reference genes for qRT-PCR in Schima Sperba were selected and their stability of gene expression under abiotic stresses (cold stress, salt stress, drought stress) and different tissues was assessed by Delt Ct, geNorm, NormFinder, BestKeeper and RefFinder. They found AP-2 and eIF-5α are most stable in Cold and Drought stress, UBQ is most stable in Salt stress. However, they didn’t provide a suitable explanation the possible reason. The similar research in Schima Sperba has been published in 2021 (Yang et al., 2021). Although in this study the authors emphasized to compare these 12 reference genes under different abiotic stress treatment, they didn’t provide abiotic stress gene expression patterns. Besides, they didn’t provide suitable description to the difference between this study and Yang’s study. The other concerns as following:
1. Many typo and formula in-consistent in this ms, they should revise them. Please check them in revised ms.
2. The title of Figure 2 should be twelve reference genes.
3. The references format should be unified. Please check all the references.

Author Response

(The authors gave the same response as above.)

Reviewer 3 Report
Authors evaluated stability of 12 putative reference genes in Schima Sperba under various conditions, which would provide a solid foundation for future gene expression analysis. I have a few minor suggestions regarding wording, grammar and figures:
* "field" should be plural
* There are two "medicine"s in the sentence
* line 32. add "of" between "efficiency" and "experimental".
* It would look better if the color legend in the first graph also match the colors of bars in the rest of graphs in Figure 4.
* line 55 and line 56. Be consistent with whether "material" is singular and plural
*line 61. "drought stress....." Check grammar of this sentence.
*line 75. Please be consistent with "qRT-PCR" or "RT-qPCR"
*In Discussion, it would be better to add more content regarding how to use the results and conclusion found in current study to answer more scientific question and guide future studies in the field.
Author Response
Dear reviewer,
Thanks very much for taking your time to review this manuscript. I really appreciate all your comments and suggestions! We have carefully considered the suggestion of reviewer and make some changes(Please see the attachment). We have tried our best to improve and made some changes in the manuscript. Please find my itemized response in below and my corrections in the re-submitted files.
Round 2
Reviewer 1 Report
In this Article, the authors reported reference genes in a tree species S. Sperba for quantitative gene expression analyses. The revised manuscript has addressed several critical issues raised in the first round peer-review. The reviewer would like to raise a few further questions before publication of this manuscript. Hope that these could be useful for improving the value of this paper.
(1) The present form of the manuscript includes all the essential information that needs to get readers’ understanding. However, the readability of could become fine by serious English-proofreading. In addition, Figures/Tables contain grammatical issues in English. A brief proofreading in English by the reviewer is shown in the attachment (only title and abstract). It would be better to conduct a native-level proofreading throughout the manuscript.
(2) The data on the expression stability for the candidate reference genes are in the core of this study while these are represented across five Figures (Fig. 3-6) and four Tables (Table 1, 3, 4, and 5). It would be better to be concise or made these compact to put on a smaller space. The important results showing the primary message should be retained on the text, and remaining results could be transferred to Supplementary materials.
(3) Ct values (Fig. 3) look useful to check the gene expression variability (expressed as “dispersion degree” in the text). The reviewer understands that UBQ (in italic) and TUB (in italic) showed the lowest and the highest dispersion degrees, respectively, among the genes tested (Line 158-160). The lowest dispersion degree might implicate the stability of UBQ. Is it right?
(4) I wonder if Fig. 6 represents the stability of combined reference genes (two-gene combination). If it is true, what would be the outcome? A combined reference gene may offer more stability than a single reference gene. If the stability of any in Fig. 6 is larger than any of the single reference genes, it would be useful and worth mentioning in the text with highlight.
Use the correct names of the tools/algorithms/methods with valid references: The references for the ΔCt and RefFinder are missing in Line 116-117. GgeNorm, NormFinder in Line 119.

Author Response
Thank you again for reviewing this manuscript so carefully, and thank you very much for all your comments and suggestions! We carefully considered your suggestions , made some changes, and tried our best to improve them, making some changes to the manuscript, hoping that our efforts would make the manuscript more standardized and scientific. Please find my itemized reply below and find my correction in the resubmitted file.

Reviewer 2 Report
1. Table 1. R2 should be R2.
2. The format of part of references are still inconsistent, please check all.
Author Response

(The authors gave the same response as above.)
